# Effects of environmental conditions and catch processing on survival probability of plaice discards in the North Sea pulse trawl fishery

**Edward Schram**[1]*, **Pieke Molenaar**[1], **Paul W. Goedhart**[2], **Jan Jaap Poos**[1,3]

**1** Wageningen Marine Research, Wageningen University & Research, IJmuiden, The Netherlands, **2** Biometris, Wageningen Plant Research, Wageningen University & Research, Wageningen, The Netherlands, **3** Aquaculture and Fisheries Group, Wageningen University & Research, Wageningen, The Netherlands

* Edward.schram@wur.nl

**Data Availability Statement:** All relevant data are within the paper and its Supporting Information files.

## Abstract

Undersized European plaice dominate the discarded fraction of the catch of the beam trawl fisheries for sole in the Southern North Sea. Effects of environmental conditions at sea and of the use of a water-filled hopper on the survival of undersized European plaice discarded by pulse trawl fisheries were explored. During trips with commercial pulse-trawlers catches were discharged in either water-filled hoppers or conventional dry hoppers. For both hoppers, undersized plaice were sampled from the sorting belt. After assessment of vitality status, sampled fish were housed in dedicated survival monitoring tanks on board. Upon return in the harbour fish were transferred to the laboratory to monitor their survival for up to 18 days post-catch. Conditions at sea, such as wave height and water temperature, as prevailing during these trips were recorded or obtained from public data sources. The overall estimate for the survival probability for plaice discarded by pulse trawl fisheries is 12% (95% CI: 8% - 18%). Both water temperature and vitality status had strong effects on survival probabilities of discarded plaice. Increasing water temperature increased mortality. The vitality of the fish could be moderately increased by using a water-filled hopper to collect the fish on deck, but we found no significant direct effect of hopper type on plaice discard survival. It seems that to increase discards survival, fish need to be landed on deck in much better condition by a reduction of the impact of capture and hauling processes on fish condition.

## Introduction

In commercial fisheries around the world, unwanted bycatches are discarded [1–3]. These bycatches are exposed to stressors during capture, handling and release. The severity of these stressors is affected by environmental conditions and characteristics of the fishery [4, 5]. Because failure to recover from these stressors ultimately results in mortality [5, 6], abiotic variables that exacerbate the severity of stressor are important for survival of discarded fish [7]. It is therefore not surprising that research trips conducted under naturally varying conditions at sea show large variation in observed discards survival [8, 9]. For conventional trawl fisheries

**Funding:** This study was funded by the European Maritime and Fisheries Fund (EMFF) grant no. 15982000049. The funders had no role in study design, data collection and analysis, decision to publish, or preparation of the manuscript. There was no additional external funding received for this study.

**Competing interests:** The authors have declared that no competing interests exist.

differences in water temperature explain part of the variation in discards survival observed in many species, including plaice and sole [10–12].

Survival of discards can also be increased by reducing the stressors capture, handling and release. Previous studies testing measures that aimed at increasing post-capture survival of fish showed promising results. Reducing haul duration from 100–130 to 60–70 minutes promoted survival of plaice discards but not sole discards [10]. Also, work on the effect of discharging catches from the cod-end into a water-filled hopper, instead of the commonly used dry hopper, suggested an increase of the survival of plaice discards [13]. The use of a water-filled hopper to increase survival of discards is of particular interest because it is relatively easy to implement while leaving the current catch-sorting process intact.

More knowledge about the effects of abiotic variables and measures to reduce mortality of fish in demersal fisheries in Europe is important in the light of the EU landing obligation [14–16]. Under the landing obligation the practise of discarding has been restricted for all quota regulated species under the Common Fisheries Policy [17]. As a result of this legislation fishers are obliged to land all undersized, damaged and marketable fish of species under quota management. This has substantial consequences for the Dutch demersal fisheries that simultaneously fish for several species including common sole (*Solea solea*) and European plaice (*Pleuronectes platessa*) [18, 19]. The mesh size of 80 mm used in the Dutch beam trawl fishery in the Southern North Sea, combined with a minimum conservation reference size of 27 cm for European plaice, results in this species dominating the discarded fraction of the catch.

Species for which high survival chances have been demonstrated when released into the sea are eligible for exemptions to the landing obligation. Despite that high survival has not been specified in the legislation to date, estimates for discards survival probabilities are required if fisheries want to apply for high survival exemptions.

Because the mortality of discarded fish may occur days after their capture, observation studies on survival of discarded fish need to observe fish often until after the end of the fishing trip [20]. Hence, these studies face logistical and technical challenges and they expensive to conduct [10]. Vitality assessment based on external damage and reflex impairment can be used as survival proxy [20–22]. To use vitality assessment as proxy, correlation between vitality and long-term survival should be established [10].

The objectives of the current study were (i) to estimate the effects of environmental conditions on discards survival of plaice caught by pulse trawlers in the North Sea, (ii) to provide estimates of survival for vitality class proxies, and (iii) to explore the effect of a water-filled hopper on discards survival of plaice. To answer these questions, we used data from 15 fishing trips research trips with commercial pulse-trawlers on discards survival and the effect of a water-filled hopper.

## Materials and methods

### Experiments

**Ethics statement.** The treatment of the fish was in accordance with the Dutch animal experimentation act, as approved by ethical committees (Experiment 2017 D0012.002). The methodology was in accordance with the guidelines for discards survival studies developed by the Workshop on Methods for Estimating Discard Survival (WKMEDS) of the International Council for the Exploration of the Sea (ICES) [23].

**Trips.** Fish for the survival experiments were collected during 15 trips on four commercial trawlers using sumwing pulse gear produced by HFK [24]. Three to six trips were made per trawler. Trips lasted between four and five days, comparable to commercial trip lengths. The trawlers were similar in size, with engine powers of 1430–1471 kW, towing two sumwings at

**Table 1. Vessel and gear specifics of the four vessels used in the study.**

| Characteristics | | Vessels | | | |
|---|---|---|---|---|---|
| | | **A** | **B** | **C** | **D** |
| Electrodes | Number | 25 | 22 | 24 | 26 |
| | Total length (m) | 6.7 | 7.5 | 7.2 | 7.4 |
| | Distance between electrodes (cm) | 42.5 | 40.0 | 42.5 | 45.0 |
| | Length electrodes on seabed (m) | 3.2 | 3.0 | 3.2 | 4.4 |
| Conductor | Number | 10 | 11 | 10 | 12 |
| elements | Diameter (mm) | 28 | 35 | 28 | 33 |
| | Length (mm) | 130 | 130 | 130 | 134 |
| | Distance between elements (mm) | 210 | 220 | 210 | 200 |
| Pulse | Power (kW/m) | 5.2 | 6.0 | 5.3 | 7.3 |
| | Width (µs) | 260 | 340 | 390 | 330 |
| | Frequency (Hz) | 80 | 60 | 45 | 60 |
| | Maximum exposure to pulse field (s) | 1.3 | 1.2 | 1.3 | 1.7 |

speeds of around 4.5 kn. The sumwings were typical for commercial gears: each 12 m wide wing deployed a 30–34-meter-long trawl, equipped with an 80–140 kg rubber disc false foot-rope and an 80mm cod-end. Peak voltages over the electrodes were 60 V for all trips. All fishery operations were conducted in the Southern North Sea according to commercial practices of the pulse-trawlers (Table 1), as described by [25, 26]. Trips were spread out over the year (Table 2) such that the potential effects of variable environmental and fishing conditions on discards survival could be studied.

**Test-fish collection.** Test-fish were randomly sampled on board during the catch-sorting process common in this fishery. In this process, catches were discharged from the cod-ends into a hopper (one hopper for each of the two cod-ends). From the hoppers, the catches were

**Table 2. Trip information and discards survival probability estimates for plaice.** Number of fish and discard survival probabilities are provided for control fish (C), test-fish sampled with dry hoppers (D), and test-fish sampled from water-filled hoppers (W).

| Trip[1] | Vessel | Year | Month | Hauls | Control (C) | | Dry hopper (D) | | Water-filled hopper (W) | |
|---|---|---|---|---|---|---|---|---|---|---|
| | | | | | N | Survival probability (%) | N | Survival probability (%) | N | Survival probability (%) |
| 1 | A | 2014 | Nov | 2 | 19 | 95 | 21 | 24 | - | - |
| 2 | B | 2015 | March | 4 | 30 | 60 | 60 | 23 | - | - |
| 3 | A | 2015 | May | 2 | 40 | 12 | 43 | 12 | - | - |
| 4 | B | 2015 | June | 5 | 30 | 17 | 51 | 4 | - | - |
| 5 | A | 2015 | July | 3 | 17 | 82 | 40 | 28 | - | - |
| 6 | B | 2015 | Sept | 4 | 55 | 31 | 47 | 4 | - | - |
| 7 | C | 2017 | May | 4 | 35 | 100 | 60 | 15 | 60 | 18 |
| 8 | B | 2017 | May | 6 | 28 | 96 | 59 | 15 | 59 | 29 |
| 9 | D | 2017 | June | 6 | 30 | 100 | 60 | 12 | 60 | 15 |
| 10 | D | 2017 | July | 6 | 30 | 90 | 57 | 4 | 59 | 10 |
| 11 | C | 2017 | Sept | 7 | 33 | 30 | 80 | 1 | - | - |
| 12 | D | 2017 | Oct | 6 | 30 | 100 | 60 | 22 | 60 | 18 |
| 13 | B | 2017 | Dec | 6 | 29 | 72 | 60 | 20 | 60 | 10 |
| 14 | C | 2018 | Jan | 6 | 29 | 72 | 57 | 18 | 60 | 12 |
| 15 | B | 2018 | Feb | 6 | 25 | 92 | 59 | 20 | 60 | 45 |

[1] Discards survival estimates for conventional dry hoppers measured in trip 1 to 6 were previously published by [10].

transported by a conveyer belt onto the sorting belt. Marketable fish were manually collected from the sorting belt by crew members. Test-fish were sampled by researchers at the end of the sorting belt, just before the point where the remaining unwanted catch, including fish with no commercial value and undersized fish, dropped into a gutter with a water supply that discharged the catch back into the sea.

For each trip, test-fish were collected from two to six hauls (Table 2). Seven out of 15 trips had a conventional dry hopper type (D) only, where 21 to 80 undersized (< 27 cm) plaice were collected per trip, from a total of 27 hauls. Each haul, equal numbers (five to ten) of fish were sampled at the start and the end of the catch-sorting process. During the other eight trips a conventional hopper and a water-filled hopper type (W) were used, and 10 fish per hopper type per haul were collected, with five fish being taken at the start of the catch-sorting process and 5 fish at the end, from a total of 46 hauls. The latter hauls thus were paired observations with identical sea conditions. The total number of observed (group) discards survival probabilities therefore equalled 119. In total, there were 1292 test-fish collected (Table 2).

**Control fish collection.** Control fish (C) were used to separate potential effects of the experimental procedures on mortality from fisheries-induced mortality [23]. These control fish were collected beforehand by commercial shrimp and pulse beam trawlers which had been requested to collect undersized plaice from short hauls. Prior to the experimental trips, control fish were stored in tanks placed in a climate-controlled room for at least three weeks. During this period, fisheries-induced mortality levelled out while surviving fish could recover from injuries and regain good condition. During storage, fish were fed once daily in the morning with live polychaete worms (*Nereis spp.)* and dead, uncooked brown shrimps (*Crangon crangon*) to visually observed satiation. Only fish in good condition, well-fed and without visible injuries, were used as control fish.

At the start of a trip, control fish were stored on deck in aerated 600L tanks. At sea the tank water was regularly renewed with surface seawater. Control fish were exposed to the exact same experimental procedures throughout the experiments as test-fish, from the moment test-fish were collected from the sorting belt. The number of control fish deployed was approximately 30% of the number of test-fish (Table 2).

**Collection of covariates.** For each haul, data on operational and environmental conditions (abiotic variables) were recorded by skippers and researchers on board or obtained from public data sources (Table 3). Handling time, being the time spent in the catch sorting process was recorded at the level of the individual fish. Tow depth was recorded by the skipper of the vessel using an echosounder. Substrate type for each haul was downloaded from EMODnet [27]. Wave height data were obtained from a database of the Dutch Ministry of Infrastructure and Water Management. Wave heights in this database were available from 14 automatic

**Table 3. Covariates and the range of their values recorded during the sea trips at haul level.**

| Abiotic variables | Range of values | Unit | Method |
|---|---|---|---|
| Fish length | 15–31 | cm | Total length of fish measured at the time of capture. |
| Tow depth | 22–50 | m | Estimated using average reading of depth sounder during haul |
| Substrate type | Mud to muddy-sand, Sand, Coarse-grained sediment | - | Downloaded from EMODnet geology seabed substrate maps using the EMODnetWFS package [27] |
| Wave height | 0.3–2.6 | m | Measured by the nearest buoy of the Dutch Directorate-General for Public Works and Water Management |
| Surface water temperature | 4.0–20.0 | ˚C | Measured in the survival units on board |
| Haul duration | 1.7–2.33 | hr | Estimated by skipper |
| Handling time | 1–34 | min | Calculated as difference between catch coming on board and putting fish in tanks. |

measuring buoys in the North Sea. Data from these buoys were linked to individual hauls used for survival, based on date, time, and nearest location. All variables were continuous except substrate type which is categorical with three categories: mud to muddy sand, sand, or coarse-grained sediment.

**Assessment of vitality status and monitoring of survival.** After collection, fish were temporarily stored in 105L plastic holding containers filled with seawater. Seawater in holding containers was regularly renewed to maintain dissolved oxygen levels during storage. Upon completion of fish collection, fish were sequentially taken from the holding containers for vitality assessment, to measure total length (TL: in cm below) and for tagging. Vitality status of each individual fish was assessed, jointly by two observers, by scoring a vitality class, with fish in class A being most vital with least injuries, and class D being most lethargic with most injuries [10]. Vitality assessment was standardized in a protocol and by joint training of the observers.

All fish were tagged with Trovan Unique glass transponders (type ID100) to allow for identification of individuals. Transponders were injected subcutaneously just behind the head using an IID100E injector. To ensure backtracking of each test-fish to its haul and hopper treatment, test-fish from different hopper treatments were stored in separate 105L holding containers prior to tagging. Tagging was completed before the next haul was sampled. Upon completion of the vitality assessment and tagging, live fish were placed in 24L tanks (see *Experimental facilities*) with a maximum of five fish per tank. Fish that were considered dead (no operculum movement for more than 15 seconds) were recorded as dead at time zero. Dead fish were not replaced by live individuals. Survival was monitored up to 18 days.

## Experimental facilities

All test-fish and control fish were housed in four custom-built monitoring units installed on board. Each unit consisted of a stainless-steel framework holding 16 24L tanks (60 cm L x 40 cm W x 12 cm H), resulting in a total capacity of 64 tanks on a vessel. Each tank was equipped with an individual water supply. A pump with a water intake approximately 2 meters below sea surface continuously supplied seawater to the tanks. Water flow rates were approximately two tank volumes per hour ($1–1.5L^{-1}$ min) to maintain proper water quality. Tanks were covered with transparent lids to limit water loss by sloshing while allowing for visual inspection of fish.

At the end of the fishing trip, the units were transported to the laboratory in a temperature-controlled truck. Transport time ranged from one to three hours depending on the home port of the vessel. During transport, each unit was placed inside a tank that was partly filled with seawater and equipped with a submerged pump to supply water to each fish tank in the unit. Fish tanks discharged their effluents in the tank in which the unit was placed, allowing for recirculation and aeration of the water.

Upon arrival at the laboratory, fish tanks were manually stacked in racks placed in a temperature-controlled room. Temperature was set at the North Sea surface water temperature at time of fish collection. All tanks were connected to a single water recirculation system consisting of a 440 L pumping tank and a 330 L trickling filter. Total system volume was approximately 3.2 m$^3$. Water in the system was continuously renewed with filtered seawater from the Eastern Scheldt at a rate of 8.6 m$^3$/d. In the laboratory, all tanks were supplied with coarse sand as bottom substrate and fish were fed daily to visually observed satiation with live polychaete worms and dead uncooked brown shrimps.

All tanks were inspected every 12 hours on board and every 24 hours after transfer to the laboratory. Tanks were inspected for mortalities by visual observation of fish movement. Dead fish were removed, but lethargic fish were not removed. Dissolved oxygen concentration and

water temperature were measured. Water flows to individual tanks were increased if oxygen saturation was below 80%.

## Statistics

Survival curves were made to plot the survival percentages of the fish over time. Then, the effects of environmental factors and hopper type on vitality status were tested using ordinal logistic regression. Finally, the effects of environmental factors, hopper type and vitality status on discard survival were tested using Generalized Linear Mixed Models (GLMMs).

**Survival curves.** For each fish that died during survival monitoring, the survival time was recorded as the time (h) since collection from catches. Survival curves presenting the development over time of survival within groups were estimated using the non-parametric Kaplan-Meier estimator [27]. Two survival curve plots were made: a first plot with the survival of control fish, and those in dry and water hoppers. A second plot was made to depict survival of the different vitality scores of the test-fish.

## Effects of covariates and hopper type on vitality status

To test the effect of environmental conditions and the hypothesis that water-filled hoppers yield better vitality status for plaice, ordinal logistic regression [28] was used. Ordinal logistic regression predicts multi-class ordered dependent variables, using the proportional odds logistic regression technique. The ordered variable was vitality class. Covariates, including handling time and environmental factors such as tow depth, surface water temperature, and wave height were used as continuous explanatory variables, while substrate type was a factorial explanatory variable. Hopper type also entered the model as a factorial effect. A random effect was added for haul, addressing the potential variability between hauls. The full model was thus:

$$Vitality = Haul\ duration + Depth + Fish\ length + Temperature + Wave\ height$$
$$+ Handling\ time + Substrate + Hopper + a_j.$$

where $a_j$ was the random effect of haul, distributed as $N(0, \sigma^2_{haul})$. The model was fit as a Cumulative Link Mixed Model [9, 29]. The random effect was tested on the full model by a one-tailed chi-square test. Subsequently, model selection based on AIC was used to determine the best model, where more complex models were only selected if $\Delta AIC > 2$.

**Discard survival.** To test for the effects of environmental factors, hopper type and vitality status on discard survival, a generalized linear mixed model (GLMM) was used on fish survival. A binomial distribution for the 0/1 (dead or alive) response was used, with a logistic link function. To account for correlations within hauls and within trips, the model was formulated as a two-way nested generalized linear mixed effects model. The potential effects of vitality class, hopper type, and surface water temperature were included as fixed effects. The model can thus be written as

$$S_{ijk} \sim B(\pi_{ijk})$$

$$E(S_{ijk}) = \pi_{ijk}$$

$$\text{logit}(\pi_{ijk}) = Intercept + Vitality + Hopper + Temperature + Substrate + Fish\ length + a_{jk}$$
$$+ b_k$$

$$a_{jk} \sim N(0, \sigma^2_{haul})$$

$$b_k \sim N(0, \sigma^2_{trip})$$

$S_{ijk}$ was coded 1 if fish $i$ from haul $j$ in trip $k$ was alive, and 0 if it was dead at the end of the experiments. All combinations of fixed effects and all one-way interactions of these fixed effects were included in a model selection procedure. Because of the large number of possible combinations of these fixed effects and their one-way interactions, the maximum number of terms tested in the model selection procedure was 5. The final model was selected based on the Akaike Information Criterion (AIC). To avoid issues with fitting the model and to aid interpretation of the results, surface water temperature and fish length entered the model after being subtracted by their respective means. The random effect was tested on the full model by a one-tailed chi-square test. Estimated marginal means (EMMs) for model factors were used as a post-hoc analysis to estimate the level of significance between hopper type or vitality class in case a significant effect was detected. The significance level was set at 5%.

Two additional models for plaice discards survival were tested. First, because of the interest in the effect of hopper type on survival, the direct effect of hopper type was tested in combination with the environmental effects found to be statistically significant in the model above. Like in the previous model, random effects of haul and trip were included. This model was applied to the subset of 46 hauls in eight trips where both hopper types were used. Second, to estimate the overall survival of plaice discards in the conventional hopper, we tested for an intercept only model, where random effects of haul and trip were included. This model was applied to the subset of 73 hauls where the conventional hopper was used, in all trips.

**Software.** Using R 4.1.2, the EMODnetWFS package [27] was used to acquire EMODnet data for substrate type. The ordinal package [29] was used for proportional odds logistic regression. The lme4 package [30] was used for generalized linear mixed models, the dredge function in the MuMIn package [31] was used for model selection. The effects package was used for effect displays of ordinal regression outputs [32, 33], the emmeans package for estimating marginal means of GLMMs [34], and the car package [33] for analysis of deviance tables.

## Results

### Survival curves

Survival of control fish was high (Fig 1A), with more than 90% of fish surviving the first 5 days after capture, and more than 70% surviving up to the end of the experiments at 18 days. Overall, most of the mortality in the experiments occurred in the first 7 days after capture. More than 80% of undersized plaice which entered the experiment in poor condition (vitality scores C and D) died within 6 days post capture. As expected, survival of fish of vitality class A was much higher at the end of the experiment: approximately 60% survived (Fig 1B). Mean survival of the test-fish in the water-filled hopper was slightly higher than test-fish in the dry hopper from the start throughout the experiment (Fig 1A).

### Effects of environmental factors and hopper type on vitality status

The ordinal logistic regression results of vitality class revealed that the random effect of haul contributed significantly to the model ($\chi^2(1) = 10.70$, p = 0.001). Subsequent model selection for fixed effects revealed that hopper type, handling time and substrate contributed to the best model. Hence, the final ordinal logistic regression model consisted of explanatory variables

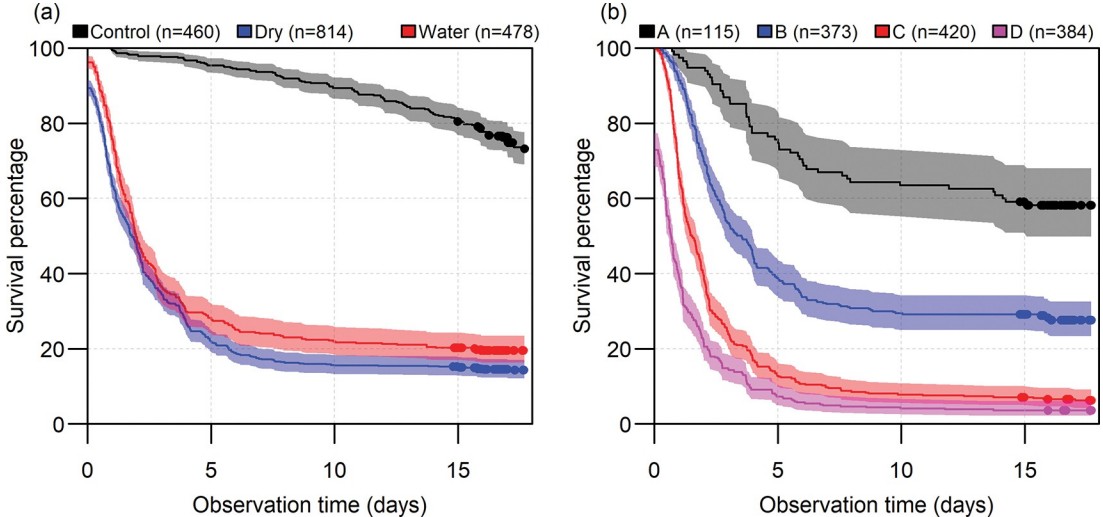

**Fig 1.** Kaplan-Meier survival curves for plaice discards by hopper type (a) and by vitality class (b). In (a), curves are plotted for control fish (Control) and test-fish for two different hopper types: Conventional dry hoppers (Dry) and water-filled hoppers (Water), where both hopper types were used in a paired design. In (b), curves are plotted for test-fish of vitality classes A to D. Note that vitality class D includes fish that were dead at the start of the experiment. Drawn lines indicate mean survival percentage over time, with shaded areas indicating 95% confidence limits. Dots indicate the end of the monitoring time for individual fish that were alive at the end of the experiments.

hopper type, handling time, and substrate, and a random effect for haul so that:

$$Vitality = Hopper\ type + Substrate + Handling\ time + a_j$$

The coefficient for the difference in the model for the water-filled hopper compared to the conventional dry hopper was -0.44 (Table 4), corresponding to a proportional odds ratio of 0.64. This means that for fish in the dry hopper treatment, the odds of being in a higher category is 1.56 times that of fish in water hopper treatments, holding constant all other variables. The effects of hopper type, substrate, and handling time can be seen in Fig 2, where the model estimated mean probability for vitality classes A and B are higher for water filled hoppers. Probabilities for vitality class D are clearly lower for water filled hoppers (Fig 2A). This means that fish coming out of the water filled hopper are generally in better condition than fish coming out of a traditional dry hopper. The effect of substrate type was such that muddy to muddy-sand substrates had higher probabilities of vitality class D, compared to sandy

**Table 4. Results from the ordinal regression analysis, testing the effects of hopper type and substrate on vitality class.**

|  | Estimate | Std. Err. | z-value | p-value |
|---|---|---|---|---|
| Threshold coefficients | | | | |
| A\|B | -3.48 | 0.46 | -7.59 | |
| B\|C | -1.58 | 0.45 | -3.52 | |
| C\|D | -0.13 | 0.45 | -0.29 | |
| Coefficients | | | | |
| Hopper: Water | -0.442 | 0.113 | -3.92 | <0.001 |
| Handling time | 0.028 | 0.009 | 3.24 | 0.001 |
| Substrate: Sand | -1.309 | 0.439 | -2.98 | 0.003 |
| Substrate: Coarse-grained Sediment | -0.917 | 0.478 | -1.92 | 0.055 |

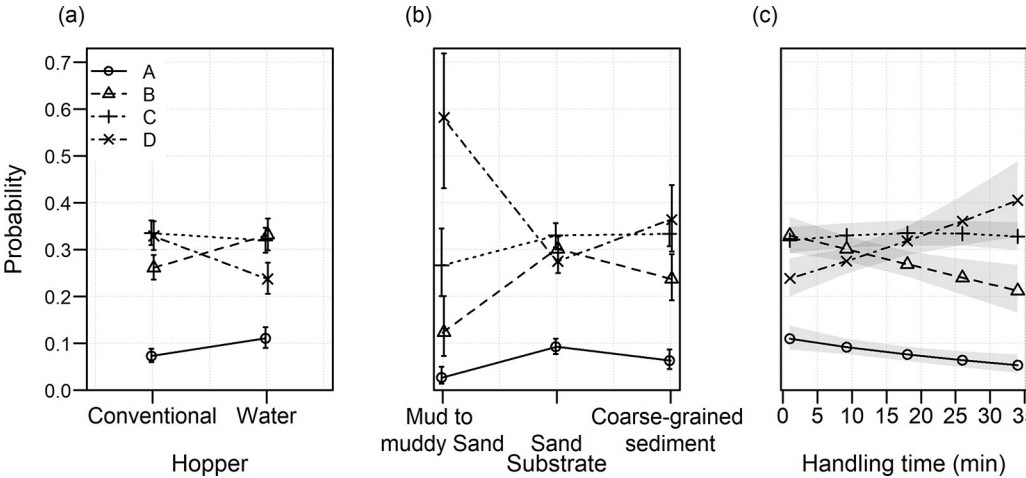

**Fig 2. Effects of hopper type, substrate and handling time on the probability of falling in each of the vitality classes.** Dots and drawn lines indicate model estimated means, with arrows and shaded areas indicating 95% confidence limits.

substrates. Sandy substrates had higher probabilities of finding fish in vitality classes A or B. Probabilities of vitality classes for the coarse-grained sediments were estimated to be in between the other two sediment types (Fig 2B). Increasing handling times of fish result in decreasing vitality: the probability of being in vitality class A or B decreases with handling time and the probability of being in vitality class D increases with handling time (Fig 2C). Finally, the standard deviation for the distribution of the random intercept effect of haul $\sigma_{haul}$ was estimated to be 0.42.

## Discards survival

The GLMM model describing discards survival that was selected based on AIC included fixed terms for vitality class, surface water temperature, and fish length, together with the one-way interaction term for vitality class and temperature. The random effect of haul and trip contributed significantly to the model ($\chi2(2) = 23.88$, p < 0.001). The final model was thus

$$S_{ijk} \sim B(\pi_{ijk})$$

$$E(S_{ijk}) = \pi_{ijk}$$

$$\text{logit}(\pi_{ijk}) = Intercept + Vitality + Temperature + Vitality*Temperature + Fish\ length + a_i + b_{ij}$$

$$a_i \sim N(0, \sigma_{trip}^2)$$

$$b_{ij} \sim N(0, \sigma_{haul}^2)$$

The best model for discard survival thus included vitality classes, while this effect of vitality class on survival depended on temperature (Table 5). The standard deviation of the random intercept effect of haul ($\sigma_{haul}$) was 0.74, and the standard deviation of the random intercept effect of trip ($\sigma_{trip}$) was 0.63. Throughout the observed temperature range, survival of vitality class A was highest, followed by B, C, and D. At the average observed temperature of 12.0°C

**Table 5. Fixed effect estimates from the final GLMM model, including standard error, z-value, and p-value.** Note that these estimates are on the logit scale.

| Effect | Estimate | Std. Error | Z-value | p-value |
|---|---|---|---|---|
| Intercept | 0.476 | 0.299 | 1.59 | 0.111 |
| Vitality class: B | -1.617 | 0.268 | -6.03 | <0.001 |
| Vitality class: C | -4.061 | 0.389 | -10.45 | <0.001 |
| Vitality class: D | -9.596 | 3.174 | -3.02 | 0.002 |
| Temperature | -0.089 | 0.064 | -1.40 | 0.162 |
| Fish length | 0.135 | 0.037 | 3.64 | <0.001 |
| Vitality class B: Temperature | -0.024 | 0.059 | -0.42 | 0.677 |
| Vitality class C: Temperature | -0.212 | 0.079 | -2.67 | 0.008 |
| Vitality class D: Temperature | -0.799 | 0.416 | -1.92 | 0.055 |

and the average fish length of 22.6 cm the survival of vitality class A was 60% (95% CI: 51%-69%) (Fig 3A). At the lowest observed temperature of 4˚C the survival of this vitality class was 75% (95% CI: 59%-87%), while at the highest observed temperature of 20˚C survival decreased to 42% (95% CI: 26%-61%). The steeper slopes of temperature-dependent survival for the other vitality classes, result in sharp drops of survival with temperature, especially for vitality classes C and D. For vitality class D survival at 4˚C was 12% (95% CI: 4%-29%), decreasing to a survival probability of less than 0.1% at 12˚C.

The survival probability within a vitality class increased with fish length (Fig 3B). For fish of the dominant vitality class (class C), this means an increase in survival from 1% to 7%.

Because vitality class depended on hopper type, discards survival depended on vitality class, and we are interested in the effect of hopper type, the model that included hopper type, fish length, temperature, and random effects of haul and trip, is of particular interest. This model was applied to the subset of data for the 46 hauls in eight trips where both hopper types were used in a paired design. This model also showed evidence for a decrease of survival with temperature and an increase of survival with fish length (Table 6). However, no significant effect of hopper type was found in this model (Table 6). The overall survival of discards for the conventional hoppers estimated in the intercept only model was 12% (95% CI: 8%-18%).

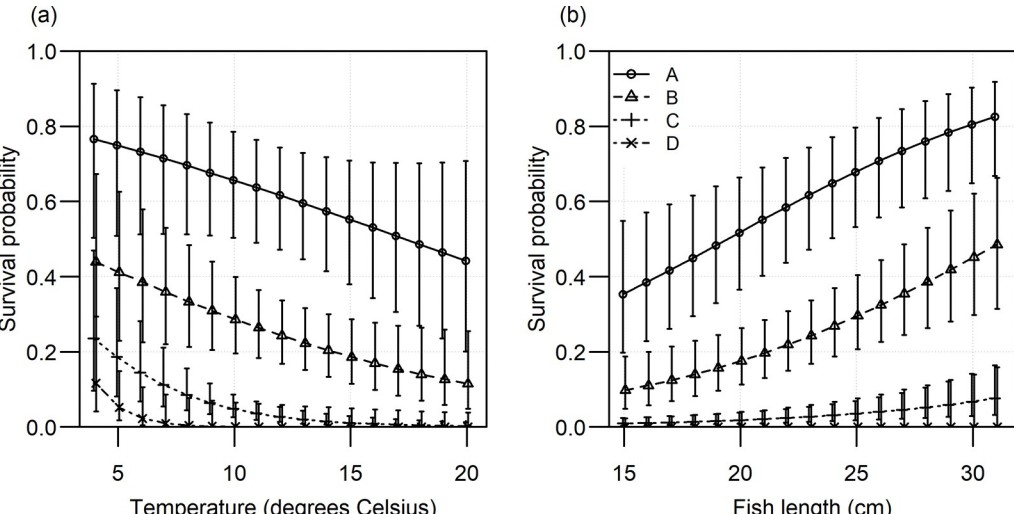

**Fig 3. Effects of fish condition (vitality class) and surface water temperature on survival probability.** Symbols indicate means, arrows indicate 95% confidence limits. Estimated effects of temperature are shown for a fish of average length (22.6 cm) and effects of fish length are shown for fish under average temperature conditions (12˚C).

**Table 6. Effect estimates from the GLMM model with hopper type, temperature, and fish length, including standard error, z-value, and p-value, for the eight trips where both hopper types were present.** Note that these estimates are on the logit scale.

| Effect | Estimate | Std. Error | Z-value | p-value |
| --- | --- | --- | --- | --- |
| Intercept | -1.896 | 0.172 | -10.95 | <0.001 |
| Hopper: Water | 0.336 | 0.177 | 1.90 | 0.057 |
| Temperature | -0.081 | 0.029 | -2.78 | 0.005 |
| Fish length | 0.068 | 0.034 | 1.99 | 0.047 |

## Discussion

The objectives of the study were to estimate the effects of environmental conditions on survival of discarded plaice and to explore the effect of a water-filled hopper on discards survival. We found that hopper type affected the vitality status of discarded fish, along with handling time and substrate. Vitality status, in turn, was a strong predictor of survival, together with water temperature and fish length.

### Experimental procedures

We used captive observation to estimate discards survival probabilities. Our test-fish were thus not actually discarded but housed in tanks to monitor their survival over time. These artificial conditions as well as the handling of the fish associated with the experimental procedures may induce additional stress and even additional mortality, in which case actual discards survival may be underestimated [26]. We employed control fish to separate potential effects of the experimental procedures on mortality from fisheries-induced mortality. Low survival of control fish occurred in four out of fifteen trips, mainly during the second week of survival monitoring. However, we consider it unlikely that this resulted in an underestimation of the survival probability due to additional non-fisheries related mortality among test-fish because most test-fish had already died before effects of experimental procedures manifested among control fish. In other words, given the difference in timing of mortality among control and test-fish, mortality among test-fish appears to be largely fisheries-induced. Captive observation may also overestimate discards survival because it excludes the contribution of predation to discards mortality [35]. Jointly taken, the discards survival probability estimates should be considered as maximum values because captivity related mortality was probably low while the captive observation excluded an unknown level of predation related mortality.

### Effects of environmental conditions

Inherent to field research in collaboration with commercial fisheries, our study did not follow a strict temporal design. Because conditions at sea cannot be controlled, a balanced experimental design with respect to variables such as wave heights or wind speeds is very hard if not impossible to achieve. The best that can be done (and was done) is to spread out trips over the year to cover as much variation in conditions at sea as possible within the logistical limitations inherent to conducting research with commercial fisheries. Consequently, this study can only provide correlational evidence and no direct causal evidence for effects of environmental conditions.

We found a negative effect of increasing surface water temperature on discard survival. This result is in line with findings by [10] who used a subset of the data. This effect of reduced survival at higher temperature was also found in other fisheries for European plaice. In otter trawl fisheries in Skagerrak, survival was significantly higher in winter (75%) than in summer (44%), with an annual surface water temperatures range of 6 to 17˚C [36]. A similar effect of

temperature was found for Danish seines in that area [37]. Likewise, a study on discarded flat-fish in the Western Baltic trawl fishery found that discard mortality ranged between 5% and 100%, with low mortalities observed in winter (4°C) and high mortalities observed in summer (14°C) [38]. These effects of sea water temperature could be related to temperature shock when bringing the fish on deck [9], but we did not test for this effect. It is also possible that in winter fish are less susceptible to hypoxia because of their lower physiological oxygen demand at lower temperatures [39].

We observed that the vitality of fish improved when using a water-filled hopper and when reducing handling time, being the time between fish coming on board and the time at which they were put back into the water. Both reduce air exposure leading to desiccation and hyp-oxia, which is an important stressor affecting plaice discard survival [37, 40, 41]. Meanwhile, the direct estimate of the increase of survival in the water-filled hopper suggests no significant effect of hopper type on survival probability.

The prevalence of the lowest vitality class (class D) was highest in the muddy to muddy-sand sediment type. On sandy sediments, the majority of observations, survival was thus higher than on muddy sediments. This finding was contrary to our expectations: we expected that abrasion of skin by sand during trawling might result in deteriorated vitality of fish, whereas softer, muddy seafloors may affect vitality less [5]. We expected sandy and coarse-grained sediments to cause mechanical injuries through increased abrasion and hence reduce survival [37, 42].

No significant effect of fish size was found on vitality status, but there was an effect of fish size on survival within vitality class. A similar positive effect of fish size on survival probability of plaice discarded by pulse trawl fisheries was reported by [43] and explained as an effect of smaller fish being more vulnerable to the overall impact of trawling. It is then surprising that fish size did not affect vitality status. Possibly the size differences in vulnerability lead to rather subtle differences in vitality that are not detected by our vitality status scoring method but do result in size dependent survival. Alternatively, it is also possible that larger fish show higher survival because they are better at copying with the conditions in captivity for survival moni-toring, either on board or in our laboratory. Indeed, a tendency for higher mortality among the smaller fish within groups kept in captivity was observed for common sole (*Solea solea*) [44]. Within groups kept in captivity, social interactions and competition for food may place smaller fish in a disadvantaged position [45]. Irrespective of the mechanism underlying the observed fish size effect on survival within vitality classes, we consider the absence of a main effect of fish size, i.e., independent from vitality class, as an artefact of the low survival for vital-ity classes C and D.

We found no effects of tow depth on plaice discards vitality or survival. Flatfish like plaice have no swimbladder, but other forms of barotrauma may occur [4]. Hence, an effect of depth has been hypothesized for discarded plaice [10, 37] and observed in at least one fishery [9]. We also found no effect of haul duration on plaice. Haul duration is expected to enhance probabil-ity of mechanical injuries through abrasion [10, 46]. A possible explanation for the absence of an effect of haul duration is the limited range of observed haul durations because research was done under commercial fishing circumstances.

## Estimates of survival for vitality class proxies

We found that the vitality classes used in this study were good estimators for discards survival with large differences in survival between classes. This makes scoring vitality status a useful tool for studying qualitive effects on discards survival probabilities of e.g., modifications to capture or catch-sorting processes prior to employing captive observation studies aimed at quantifying these effects [40, 43, 47].

While the survival of discarded fish in good condition may be high, their contribution to the overall discards survival was small as the proportion of fish in good condition in the catches was often low. In fact, most fish (56% for water-filled hoppers and 66% for dry hoppers) in the catches were in poor condition (vitality classes C and D), with survival < 10%. The water-filled hopper prevented further deterioration of fish condition during cod-end discharging and catch processing. While the use of the water-filled hopper led to a slight shift towards more vital plaice discards, the proportion of fish in good condition (score A) increased only from 7% to 11%. This small shift towards better fish condition did not lead to a statistically significant increase of discards survival. The failure to detect an effect of the water-filled hopper could be explained by survival being largely determined by capture and hauling processes prior to discharging the fish in hoppers. When improved fish handling on deck does not undo damage already done in the trawl, a water-filled hopper or any other measure aimed at preventing deterioration of fish condition will have only a limited positive effect on survival probability. We therefore recommend that future studies into increasing discards survival probabilities focus on avoiding unwanted bycatches of undersized plaice, i.e., improve the size selectivity of the gear and on increasing the proportion of fish in good condition in the catches by reducing the impact of capture and hauling processes on fish condition. Once the condition in which the fish are landed on deck is improved, it is possible that the contribution of a water filled hopper to survival increases.

## Conclusions

Both water temperature and vitality status of fish had strong effects on survival probabilities of discarded plaice. Increasing water temperature increased mortality. Moreover, the vitality of the fish could be moderately increased by using a water-filled hopper to collect the fish on deck. However, we did not detect a statistically significant effect of the water-filled hopper in the 46 hauls where these were tested. Overall estimate for the survival probability for plaice discarded by pulse trawl fisheries was 12% (95% CI: 8% - 18%). We recommend that future studies into reducing fisheries mortality among undersized plaice focus on avoiding their catch and on reducing the impact of capture and hauling processes on the catch.

## Supporting information

**S1 Data.**
(CSV)

## Acknowledgments

The authors thank the following persons and organisations for their contributions to this study: the skippers and crews of the fishing vessels UK33, TX3, GO23, GO31, TH10 and OD3, Pim van Dalen, Ainhoa Blanco, Ad van Gool, Emiel Brummelhuis, Yoeri van Es, Ewout Blom, Joe Freijser, Raoul Kleppe and Pim Boute. We thank Van Wijk installaties en constructies BV, Maaskant Shipyards Stellendam BV and Visserij Coöperatie Urk (VCU) for preparing and installing the technical installations at the vessels. Finally, we thank two anonymous reviewers and the editor for their constructive comments that helped improve this manuscript.

## Author Contributions

**Conceptualization:** Edward Schram, Pieke Molenaar.

**Data curation:** Edward Schram, Pieke Molenaar, Paul W. Goedhart, Jan Jaap Poos.

**Formal analysis:** Pieke Molenaar, Paul W. Goedhart, Jan Jaap Poos.

**Funding acquisition:** Edward Schram, Pieke Molenaar.

**Investigation:** Edward Schram, Pieke Molenaar.

**Methodology:** Edward Schram, Pieke Molenaar.

**Project administration:** Edward Schram.

**Writing – original draft:** Edward Schram, Pieke Molenaar, Paul W. Goedhart, Jan Jaap Poos.

**Writing – review & editing:** Edward Schram, Jan Jaap Poos.

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
