## [Decision Letter · Decision Letter 0]

24 Oct 2022

PONE-D-22-20005Increasing survival probability of plaice discards in the North Sea pulse trawl fisheryPLOS ONE

Dear Dr. Schram,

Thank you for submitting your manuscript to PLOS ONE. I now have in hand two thorough reviews of your manuscript. Although one of the reviewers recommended rejection, I have decided to give you a chance to carry out a (major) revision based on their comments (pasted below).

If you decide to revise and resubmit your manuscript to PLOS ONE, please carefully consider all the reviewers' comments and provide a revision letter in which you explain how you have dealt with each of them.

We look forward to receiving your revised manuscript.

Kind regards,

Even Moland

Academic Editor

PLOS ONE

Journal Requirements:

This study was partly funded by the European Maritime and Fisheries Fund (EMFF) grant no. 15982000049. 

This study was partly funded by the European Maritime and Fisheries Fund (EMFF) grant no. 15982000049. 

This study was partly funded by the European Maritime and Fisheries Fund (EMFF) grant no. 15982000049. 

However, funding information should not appear in the Acknowledgments section or other areas of your manuscript. We will only publish funding information present in the Funding Statement section of the online submission form. 

This study was partly funded by the European Maritime and Fisheries Fund (EMFF) grant no. 15982000049. 

6. We noted in your submission details that a portion of your manuscript may have been presented or published elsewhere. Please clarify whether this publication was peer-reviewed and formally published. If this work was previously peer-reviewed and published, in the cover letter please provide the reason that this work does not constitute dual publication and should be included in the current manuscript.

7. Your ethics statement should only appear in the Methods section of your manuscript. If your ethics statement is written in any section besides the Methods, please move it to the Methods section and delete it from any other section. Please ensure that your ethics statement is included in your manuscript, as the ethics statement entered into the online submission form will not be published alongside your manuscript. 

Reviewers' comments:

Reviewer's Responses to Questions

**Comments to the Author**

1. Is the manuscript technically sound, and do the data support the conclusions?

Reviewer #1: Partly

Reviewer #2: Partly

2. Has the statistical analysis been performed appropriately and rigorously? 

Reviewer #1: I Don't Know

Reviewer #2: No

3. Have the authors made all data underlying the findings in their manuscript fully available?

Reviewer #1: No

Reviewer #2: Yes

4. Is the manuscript presented in an intelligible fashion and written in standard English?

Reviewer #1: Yes

Reviewer #2: Yes

5. Review Comments to the Author

Reviewer #1: Summary:

Schram et al. collected undersized plaice caught in the pulse trawl fishery and monitored their survival to 18 days under varying conditions. They found better fish condition and higher survival probability in lower water temperature and by using water-filled hoppers instead of the conventionally used dry hoppers. They also estimate effects of fish length, substrate, tow depth and handling time on fish condition and survival.

Overview:

Overall, I think this was a well-performed experiment to document survival probabilities of discarded plaice. The fish collection and experimental procedures are well-described. The estimated relationships between fish condition at catch and survival probability are useful, since it is much easier to record fish condition than do lab experiments to measure survival. The other results of the effects of substrate, fish length, water temperature, and tow depth were also interesting. However, I have a couple non-trivial concerns with the main result (as stated in abstract and emphasized in title) that the water-filled hopper increases survival. Once the authors address these, or explain why it is not necessary to, I would recommend publication.

Main concerns:

1. Uncertainty estimates, i.e. 95% CIs, for the 12% (dry) and 16% (water-filled) should be added. The effect of water-filled hopper is only marginally significant in the final model (p = 0.027), so the emphasis on this result feels overstated.

2. The result also seems overstated because I would characterize both 88% and 84% mortality as high—and the true mortality of releasing fish at sea may (probably?) be even higher. There is no discussion of how survival in a lab tank after being caught differs from surviving being thrown back into the sea. My guess is that lab survival would be an overestimate, but I can also imagine reasons why survival back in the sea could be higher. I think at least acknowledging this uncertainty in your discussion is warranted.

3. Many of the trips with low control survival used dry hoppers but not water-filled hoppers, and it is not clear which data were used in the GLMM analyses. Sorry if I should see this, but it is not clear to me after multiple readings. At first, I assumed the control fish were included and represented by the intercept, which would inform the model about the haul/trip random effects vs. the dry hopper effect for hauls without water-filled hopper data. But I don’t see a “Hopper: Dry” effect in Tables 4 and 6. Were all the dry and water-filled hopper data included, but not the control? This seems important because there are several hauls with low survival of control and dry hopper fish that did not have water-filled hopper fish. I think the most proper handling of these data would be to only include hauls with both dry and water-filled hopper fish in the GLMMs. This seems more in line with how the experiment is portrayed in the methods, twin trawls with one cod end going into each hopper type, a paired design to test the effect of hopper type. An alternative could be to remove hauls with low, e.g. 50%, survival of control fish. I understand the random effects of trip and haul may somewhat mitigate this, but it doesn’t seem sufficient.

4. There is no data availability statement and raw data are not included in the article.

Line-specific comments:

25 uncertainty estimates should be provided for the 12% and 16% effects, especially as they are the main results

36 would be useful to have rough numbers instead of “part”

96 should this be 117 total hauls (27 + 45 + 45)?

189-229 please clarify which data were used in the GLMM analyses (long comment #3 above)

277 calling this the “final GLMM” is confusing because you have one more model below

301-307 this model is not described in the methods or here. Since it is a subset of the model on line 216, and similar to 283, perhaps describing how it differs is enough (i.e. exchange ‘hopper type’ for ‘vitality’, and no interaction). Clarify here that random effects of haul and trip were included.

307 again, uncertainty estimates should be provided for the 12% and 16% effects

394-401 I don’t think it is fair to say “The limited effect of the water-filled hopper can be explained” or “it is likely that the contribution… increases” here. This paragraph lays out a reasonable hypothesis, but this wording seems too strong. Suggest “could be explained” on 394 and “it is possible” on 401.

Tables 5-6 please include the random effect standard deviation estimates + SE, these are fixed effects in the GLMMs too.

Table 5 why was the control survival so low in Trips 3, 4, 6, and 11 (12-31%, whereas most in 2017-2018 are 72-100%)? And why were none placed in the water-filled hopper? Could add to discussion.

Reviewer #2: Evaluation of PONE-D-22-20005

This study assesses the survival probability of undersized plaice caught by pulse beam trawls in the North Sea.

Generally, survival of discards from trawl fisheries is influenced by numerous factors. Capture stressors include factors like net entrainment, crushing, wounding, sustained swimming until exhaustion, catch size, catch composition and interactions with other species in the catch. Fishing conditions include gear specifications, towing time and speed, light conditions, water and air temperature, hypoxia/anoxia, sea conditions, fish size, species or deck handling. Biological attributes are also important (Revill 2012). The real conditions a discarded fish experiences in the wild after the discard happens are, however, rarely considered and sophisticated experimental setups have been developed (and agreed upon by ICES to be considered relevant) to assess discard survival under controlled, though unrealistic conditions – like this study.

Moreover, survival studies determined by one study are inherently difficult to reproduce because so many factors differ and change each time you conduct such an experiment. It is not uncommon that one studies identifies a factor that cannot be found in another study. Hence, absolute values are relatively uncertain while relative differences between treatments groups within a study may contain useful information. Therefore, I think that the standard discard survival studies, like the one presented here, are relatively unprecise experiments and can only provide limited and rarely new and novel insights into the real processes deciding whether or not a discarded fish will survive, i.e. the individual mechanisms usually remain hidden behind some survival probabilities.

The authors use a title which indicates that they think the difference between 12 and 16% survival is “moderately” high. Well, 16>12, but the difference is rather low. Would another study really show the same difference (reproducibility)? Therefore, isn´t the more appropriate (and interesting) message rather the other way around: surprisingly little effect of water-filled hopper…?! If the hopper effect is the main focus of the study, it would have been interesting to a specific experiment set on top of these fish to look deeper into the processes occurring during the hopper process, e.g. using physiological measurements.

Unfortunately, such a study is a laborious and expensive endeavor but there are several shortcomings related to this study. In the specific comments below I list e.g. 15 trips spread over 4 years; several factors such as year, vessel, species composition or area that likely to contribute to the differences, were not considered in the analysis. Another point is that all experimental fish were tagged but tagging mortality was not separately assessed. If e.g. tolerance against tagging differs between seasons, it could have been a confounding factor in the analysis. Moreover, samples for this study were collected from 2014 to 2018. In the meantime, pulse beam trawling is no longer allowed in EU waters, mainly due to “successful” campaigning of environmental NGOs. So presently, the results are not relevant for any fishery because such a fishery does no longer exist. Pulse beam trawling, however, has a number of obvious advantages and the promising possibilities of this gear type have not even been fully explored before it was banned. The ban does, however, not mean that pulse beam trawls will never come back (especially given e.g. the recent rise in gasoline prices), but the authors provide information on this issue. Moreover, they also don’t discuss if and how the results from pulse beam trawlers (which no longer exist) can be extrapolated to common beam trawlers. There are no direct comparisons to results from common beam trawlers in the ms.

Not sure if this study is worth being published in PlosOne. There are so many limitations linked to this study and given its design, it can only provide correlational evidence, no causal evidence. Therefore, unfortunately the overall scientific progress in understanding discard survival dynamics is minor so that my first recommendation for this ms is rejection. Alternatively, if considered worthwhile to publish the surprisingly minor hopper effect, a major revision could be considered.

Specific comments

Title: see comment above

Abstract: 1 or 2 introductory sentences into the research topic/question would be welcome. This should be taken up in the last sentence of the abstract.

Conditions at sea: quite unspecific – why not mention (some key) metrics considered?

L 15: … undersized European plaice discarded… The fish were not discarded but selected, transported etc and kept in tanks. The wording is suggestive.

L 16: 15 trips: the fact that these trips were carried out during 4 different years is a major shortcoming of the study. It is not clear why pulse beam trawlers were used.

L 18: fish condition: could be misunderstood e.g. with Fulton´s K (fish body condition).

L 25: is the difference between 12 and 16% significant?

If on average 88 or 84% of the undersized plaice die, shouldn´t measures others than changing to a water-filled hopper be considered? It seems to be taken for granted that so many fish have to die. Maybe the fishery has to change, irrespective of whether a water-filled hopper can reduce mortality by 4%...? If the conclusion really to recommend water-filled hoppers and maybe not, to critically re-consider the way this fishery is conducted and think if there are no better measures to reduce the discard (e.g. by gear modifications)?

For me the most important/robust findings of this study are only “hopper has no effect”, and the usefulness of the vitality class (published in van der Reijden et al 2017. already).

Keyword: discard survival; “abiotic variables” doesn´t seem to be a useful term here – anything more specific? Or instead North Sea or something related to the method? Pleuronectes platessa?

Introduction

Several of the issues mentioned in this review were not appropriately touched upon in this Introduction, e.g. large variability in discard studies, correlational vs causal relations, multi-species catches but focus on plaice (why only one species? Why plaice?), why pulse beam trawlers, why should a water-filled hopper be the solution to the high discard mortalities?…)

M&M

It seems that the factors Vessel, Area, and Year were not accounted for in the analysis. And apparently there was no pairwise testing of vessel-trip on the treatment hopper.

L 77: In the study 15 trips on 4 commercial trawlers were used. The factor vessel (inter-vessel variation) is not accounted for in the analysis despite the fact that vessel (closely related to the specific gear and the performance of the captain with his ship on a particular fishing ground) is often a major source of variation.

The sampling covered the time period 2014-2018, with no sampling in 2016, i.e. 15 hauls from 4 trawlers from 4 years. However, there are no samples from April or August. The factor year is not accounted for the analysis (inter-year variation). Moreover, different year classes of plaice must have been caught in different years. Stronger or weaker year classes may bring fish of different fitness/condition factor; this was not accounted for in the analysis.

It seems that the sampling did not follow a strict temporal design, so that many outside (environmental) factors could not be controlled and are assumed to be accounted for by the more-or-less randomness of how the samples were taken.

L 90: Seven out of 15 trips had…

L 96: … 27 hauls. During the other 8 trips (and a total of 20 hauls), both a … were used, and 10 fish per …

I couldn´t find information on the duration of the hauls (range, mean, SD), and if the fishery was conducted at day and/or night or 24 hs. Time of day can influence the catch composition and hence the survival probability.

It is unclear what the range and mean of the total catch weight per haul was. The species composition of the hauls is not given and was not accounted for in the analysis (e.g. round fish (with gas bladders) can make the cod end float; flounder or other organisms with a rough surface can hurt the skin and mucosa of plaice).

L 89: How did the authors ensure that a single fish could be backtracked to a specific haul and hopper when the fish were kept in tanks prior to tagging?

L 90: For each of the 15 trips, test-fish…

Unclear how the fish were selected. For instance, please clarify if each time a balanced number of fish was selected. How did you chose when to pick which fish?

L 94: where do the 45 hols come from?

L 96: Please clarify: x+y+z = 119 What is x,y,z? Reference to Table 2 at the end of the paragraph could be useful.

L 97: ... were sampled onboard during …

L 106: assess instead of separate?

L 111: fish were fed once (?) daily with… At what time?

L 126: Substrate type for each haul was …

L 129: used for survival – syntax unclear

L 134: in 105L plastic (?) holding …

L 137: Tagging, especially in smaller fish, could have caused an additional source of mortality and an additional source of variation (if e.g. tolerance against tagging differs between seasons) which was not assessed in the experiments.

L 138: Fish condition could also be understood as fish body condition, such as Fulton´s condition factor. Maybe use more unambiguous terminology

L 139: Were different persons involved in the assessment? How was their performance compared between each other? And was the factor “person/assessor” accounted for in the analysis?

The experimental facilities deprived the fish from any natural predation (e.g. sea gulls at the water surface, and any other aquatic vertebrate or invertebrate predators in the sea), from parasites and bacteria, viruses and other species interactions that a flatfish would encounter at the sea floor post-discard and which all are likely to influence and further increase the discard mortality. But the artificial monitoring units used in this study could also lead to unnatural mortality because this may also have caused additional stress. Hence the results are obviously not directly transferrable to conditions in the wild and are biased. How relevant are the results for real life? One could even say that this is just an experiment and that the results are not transferable to real life. However, a discussion of these shortcomings is presently missing in the ms.

L 168: .. with dead or alive polychaetes?

L 200: The model does not account for several possible effects such as vessel, fishing ground, year or presence/absence of other species in the catches or catch composition.

To me it is not fully clear how data of 8 trips with both hopper types (identical sea conditions) were analysed in comparison to the 7 trips with a conventional dry hopper.

L 211: and within trips, the model …

L 212: Is SST considered equal to bottom water temperature, i.e. the habitat where plaice usually live?

L 222/223: Here the authors state that the design contains so many possible interactions…

L 229: is fiducial the right term? Significance level was set at 5%.

Results

L 242: revealed… was relatively high. The use of the term “test-fish” and “control fish” could be confusing.

L 248: Mean survival of the fish : of which fish? Test.fish, or control fish or plaice?

In Fig. 1 I see a continuous decline for control vs a drop at the beginning and plateau after ~7 days for the other fish

L 273: Finally, the standard …

L 274: and was does this mean then?

L 278, 279: maybe consider to use SST instead of temperature.

L 294: temperature-dependent

L 295: … was only 12%

L 296: negligible: please say how high or low this was.

L 299: …/(class C), it was linked to an increase in survival from 1% to 7% This is still very low an mean that 99 alias 93% died!

L 301-303: wording unclear to me. Vitality class was depended on hopper type? Was there a bias in the fish that came from the different hopper types?

L 303, 304: showed, not shows

L 306: how useful / relevant is a mean temperature?

L 30/: or mortality decreased from 88 to 84%. What is the confidence interval around these estimates? Given the number of factors potentially playing a role in influencing survival, is this really considered a difference that is stable and reproducible and worthwhile to be called a difference? Maybe the message is rather the other way around: surprisingly little effect of hopper type and an overall high mortality of experimental plaice. The whole aspect of the use of alternative fishing gears that can reduce the bycatch before it is hauled onboard is not discussed at all. Certainly, fishing gear which avoids the capture of non-targeted fish sizes under water and during the capture process are much more likely to reduce bycatch and hence survival of fish than hopper type. This may have to be mentioned in the discussion. Presently, the reflections on alternative measures are rather shallow.

Discussion

L 32=: with an annual SST range of 6 to 17 °C

L 323: and higher mortalities observed in summer (temperature range: x-y °C) – and delete next sentence

L 327: do the fish experience hypoxia on deck? They rather suffocate because their gills don’t work outside water. Hypoxia operates when the O2 level of the water is very low, but when the fish are out of the water, hypoxia can´t apply. The reasons for higher survival in winter may be more complex and may involve factors such as physiological/hormonal state of the fish (poikilotherms), the characteristics of the mucosa, reduced metabolism, less irradiation on deck, lower temperature difference between sea floor and deck etc. Again, many (coinciding) factors are involved. This study can only provide correlational evidence, no causal evidence. This should be acknowledged in the scientific argumentation and in the wording.

L 329: condition improved when using a water-filled hopper – can this really be 100% attributed to the hopper type? Was it significant? Did any extreme data drive the difference (e.g. a few hauls with very high or low survival in either of the hopper types)? The difference is really small.

L 333: moderate? 4 % is rather minor, Or: a slight increase. I think if the authors want to follow their line of argumentation that they think that 4% is “moderate”, a table is needed showing that the difference was really consistent. My feeling is that the message is rather: hopper type has a surprisingly small effect. And then it would be really interesting to read/see how the authors argue for an “almost no-effect” outcome because their “ideological pair of glasses” prior to this study most likely was that hopper type must have an effect. The densities in the hopper are usually very high, and the fish likely have difficulties to ventilate the operculum when so many other fish are around, lots of mucus is released and the fish may suffocate anyway, not matter if water is supplied.

At no point in the Discussion do the authors explain why the study was conducted on pulse beam trawlers and not on standard beam trawlers. What was the particular benefit or reason for using a pulse beam trawler to assess the role of a hopper effect?

L 336: … lower water temperature…

L 338: … higher temperatures…

L 350: did catches from sandy grounds have more roundfish? Their swimbladders may cause the cod end to float, see e.g. Kraak et al. 2018. Or was there a vessel effect? Any additional helpful information coming from the data themselves?

L 365: Could the paragraph be shortened? It mainly contains speculations but does little to advance our understanding about underlaying mechanisms for discard mortality. For me, this paragraph is a nice example that such studies can only provide some superficial correlational evidence, but they lack the power to understand underlaying/causal mechanisms, because 1) there are so many factors involved, 2) the fate of individuals is not considered as the focus is on integrated group means and averages.

L 373: … fishing reflected commercial conditions Or settings.

L 376: … study were…

L 382: … be higher, their ….survival was…catches was…

6. PLOS authors have the option to publish the peer review history of their article (what does this mean?). If published, this will include your full peer review and any attached files.

Reviewer #1: No

Reviewer #2: No

---

## [Author Response · Author response to Decision Letter 0]

19 Apr 2023

Please refer to the attached file "Response to reviewers" for our detailed response to all editor and reviewer comments.

---

## [Decision Letter · Decision Letter 1]

23 May 2023

PONE-D-22-20005R1Effects of environmental conditions and catch processing on survival probability of plaice discards in the North Sea pulse trawl fisheryPLOS ONE

Dear Dr. Schram,

Thank you for submitting your manuscript to PLOS ONE. After careful consideration, we feel that it has merit but does not fully meet PLOS ONE’s publication criteria as it currently stands. Therefore, we invite you to submit a revised version of the manuscript that addresses the points raised during the review process.

I share reviewer 1's opinion that your revision has adequately addressed the comments provided in the first round of reviews. Additional comments are provided by reviewer 1. We look forward to receiving your final revision addressing the minor additional comments provided below.  

We look forward to receiving your revised manuscript.

Kind regards,

Even Moland

Academic Editor

PLOS ONE

Journal Requirements:

Additional Editor Comments:

Dear authors,

I share reviewer 1's opinion that your revision has adequately addressed the comments provided in the first round of reviews.

Please provide a revised manuscript addressing the additional comments provided by reviewer 1 at your earliest convenience.

Reviewers' comments:

Reviewer's Responses to Questions

**Comments to the Author**

1. If the authors have adequately addressed your comments raised in a previous round of review and you feel that this manuscript is now acceptable for publication, you may indicate that here to bypass the “Comments to the Author” section, enter your conflict of interest statement in the “Confidential to Editor” section, and submit your "Accept" recommendation.

Reviewer #1: (No Response)

2. Is the manuscript technically sound, and do the data support the conclusions?

Reviewer #1: Yes

3. Has the statistical analysis been performed appropriately and rigorously? 

Reviewer #1: Yes

4. Have the authors made all data underlying the findings in their manuscript fully available?

Reviewer #1: Yes

5. Is the manuscript presented in an intelligible fashion and written in standard English?

Reviewer #1: Yes

6. Review Comments to the Author

Reviewer #1: The authors have adequately addressed my primary concerns from the previous version and I do not see further issues with publication. Below I list some final suggestions for consideration.

22-23 this sounds like you had both dry and water-filled hoppers on 15 trips, but there were only 8 trips with water-filled hoppers. Could change to ‘8’, since this is the sample size used in the GLMM, or perhaps remove the number, just say ‘During trips with …’

35 cut “fish”

65 is “high survival” defined with a number/range? If so, please include. In either case, it would be nice if you can clarify what, specifically, fisheries need to document to obtain an exemption.

91 Trips

285-286 suggest ‘where both hopper types were used in a paired design.’

528-529 The water-filled hopper effect was pretty close to significant, p = 0.057, so I suggest changing this and adding the estimate of survival in the water-filled hopper treatment. Something like, ‘Overall, the estimate for the survival probability for plaice discarded by pulse trawl fisheries was low (12% with 95% CI: 8% - 18%). The water-filled hopper had a weak positive effect on survival (p = 0.057, 16% with 95% CI xx%-yy%).’ And then a final sentence with your recommendation from previous paragraph, a summary of ‘We therefore recommend that future studies into increasing discards survival probabilities focus on avoiding unwanted bycatches of undersized plaice, i.e., improve the size selectivity of the gear and on increasing the proportion of fish in good condition in the catches by reducing the impact of capture and hauling processes on fish condition.’

7. PLOS authors have the option to publish the peer review history of their article (what does this mean?). If published, this will include your full peer review and any attached files.

Reviewer #1: No

---

## [Author Response · Author response to Decision Letter 1]

26 May 2023

Please refer to the enclosed file "Response to reviewers".

---

## [Editor Report · Decision Letter 2]

29 May 2023

Effects of environmental conditions and catch processing on survival probability of plaice discards in the North Sea pulse trawl fishery

PONE-D-22-20005R2

Dear Dr. Schram,

We’re pleased to inform you that your manuscript has been judged scientifically suitable for publication and will be formally accepted for publication once it meets all outstanding technical requirements.

Kind regards,

Even Moland

Academic Editor

PLOS ONE
---

## [Editor Report · Acceptance letter]

2 Jun 2023

PONE-D-22-20005R2 

Effects of environmental conditions and catch processing on survival probability of plaice discards in the North Sea pulse trawl fishery 

Dear Dr. Schram:

I'm pleased to inform you that your manuscript has been deemed suitable for publication in PLOS ONE. Congratulations! Your manuscript is now with our production department. 

Kind regards, 

on behalf of

Dr. Even Moland 

Academic Editor

PLOS ONE